# Solution-state methyl NMR spectroscopy of large non-deuterated proteins enabled by deep neural networks

Gogulan Karunanithy [1], Vaibhav Kumar Shukla [1,2] & D. Flemming Hansen [1,2] ✉

Methyl-TROSY nuclear magnetic resonance (NMR) spectroscopy is a powerful technique for characterising large biomolecules in solution. However, preparing samples for these experiments is demanding and entails deuteration, limiting its use. Here we demonstrate that NMR spectra recorded on protonated, uniformly [13]C labelled samples can be processed using deep neural networks to yield spectra that are of similar quality to typical deuterated methyl-TROSY spectra, potentially providing information for proteins that cannot be produced in bacterial systems. We validate the methodology experimentally on three proteins with molecular weights in the range 42–360 kDa. We further demonstrate the applicability of our methodology to 3D NOESY spectra of *Escherichia coli* Malate Synthase G (81 kDa), where observed NOE cross-peaks are in good agreement with the available structure. The method represents an advance in the field of using deep learning to analyse complex magnetic resonance data and could have an impact on the study of large biomolecules in years to come.

Nuclear Magnetic Resonance (NMR) spectroscopy is a ubiquitous technique in material science, chemistry, structural biology, and clinical diagnosis. In the biosciences, NMR provides unprecedented insight into functional motions and non-covalent interactions with atomic-level resolution. However, NMR is notoriously insensitive, so maximising resolution and sensitivity is a perpetual challenge within all areas of NMR spectroscopy. Nuclear spin-relaxation, the process by which equilibrium magnetisation is restored and detectable NMR signal is lost, scales rapidly with molecular size, making it challenging to study large biomolecular systems by solution-state NMR. This has meant that individual NMR experiments are traditionally associated with size limits, above which most signals are broadened beyond detection.

Over many decades, a series of developments have raised the size-limits of detection for biomolecular NMR applications, combining advances in hardware, sample preparation and pulse sequence development. The introduction of methyl-TROSY methods[1], wherein methyl-bearing side chains are used to probe biomolecular structure and dynamics, provided a step-change in molecular weight limitations for solution-state biomolecular NMR and now routinely allow applications to systems of several hundreds of kDa. Using these techniques makes it possible to study systems up to the megadalton molecular weight range. A key requirement, however, for attaining high quality methyl-TROSY spectra is that the protein should be prepared with a very high level of deuteration. Consequently, in practice, for methyl-TROSY NMR studies, the proteins produced are completely deuterated with the exception of [[1]H, [13]C] labelled methyl moieties in specific side chains, typically those in isoleucine, leucine, methionine, and valine. There are now well-established protocols to introduce such labels, which rely on the addition of specifically labelled precursor compounds to minimal and deuterated media[2,3]. However, this uniform deuteration has several disadvantages, including extra costs and typically lower yields of expressed protein. Furthermore, deuteration is not possible for many systems of considerable biological interest,

[1]Department of Structural and Molecular Biology, Division of Biosciences, University College London, London WC1E 6BT, UK. [2]The Francis Crick Institute, London NW1 1BF, UK. ✉e-mail: d.hansen@ucl.ac.uk

including proteins that can only be expressed in mammalian systems. As such, the ability to obtain high-quality $^{13}$C-$^{1}$H correlation maps from uniformly $^{13}$C labelled protonated proteins is highly desirable. The uniform labelling is easier, cheaper, and gives access to peaks associated with all methyl-bearing side chains rather than just those where the appropriate precursor has been added during protein expression. Importantly, such a method would also avoid the need for deuteration and pave the way for characterisations of large proteins that can only be expressed in mammalian systems.

In this era of burgeoning applications and developments in AI, from computational structural biology[4,5] to sophisticated large language models[6], it is natural to look for solutions within this field for the challenges encountered in characterising large proteins. In this context, we and others have recently demonstrated that deep neural networks (DNNs) can be trained to accurately transform[7–9] and analyse[10–12] complex NMR data. The most recent applications use supervised deep learning, where a DNN is supplied with an input and a target training dataset and through a training process the DNN attempts to determine the mapping between the two. Typically, this training requires very large amounts of training data, but importantly, as has been noted in several prior studies[8,13,14], it is possible to simulate an arbitrary amount of realistic training data for magnetic resonance based supervised deep learning, avoiding a significant potential data bottleneck. Deep learning methods have now been successfully applied to several tasks in magnetic resonance spectroscopy including the analysis of DEER data[13], reconstruction of non-uniformly sampled spectra[7,9,14], peak-picking[11,15], and virtual homonuclear decoupling[8].

In this study, we demonstrate that deep neural networks (DNNs) can be used, in conjunction with the traditional Fourier transform[16], to deliver very high-quality $^{13}$C-$^{1}$H correlation spectra from uniformly $^{13}$C protonated samples, including large proteins whose size limits have traditionally rendered them inaccessible to NMR. The DNNs presented below are trained to map the broad $^{13}$C-$^{1}$H spectra of uniformly labelled protonated samples to spectra that are akin to classical methyl-TROSY spectra. This is achieved by removing the effect of one bond $^{13}$C-$^{13}$C scalar couplings and increasing the resolution of both the $^{1}$H and $^{13}$C dimensions by effectively sharpening the observed cross-peaks (Fig. 1). We robustly assess the trained DNNs on synthetic data and show the applicability of the trained DNNs on experimental data for proteins with increasing size: HDAC8 (42 kDa), MSG, (81 kDa), and α7α7 (360 kDa). Finally, we extend the method to obtain 3D Methyl NOESY NMR spectra of MSG, which can aid in chemical shift assignments and/or structural characterisations.

## Results

Attempting to obtain high-quality $^{13}$C-$^{1}$H correlation maps on large proteins using classical approaches, such as $^{13}$C-$^{1}$H HSQC spectra, is hindered by several factors. Firstly, since the proteins are uniformly $^{13}$C labelled they will be subject to one-bond $^{13}$C-$^{13}$C scalar couplings that will evolve during indirect chemical shift evolutions, and split signals into multiplets and thus complicate interpretation of the spectrum. Of perhaps even greater significance, is the lack of deuteration in the system, which will lead to substantial line-broadenings in both the $^{13}$C and $^{1}$H dimensions as a result of significantly increased dipolar relaxation. Consequently, peaks in the spectra will be very difficult to identify and difficult to assign to specific sites in the protein, making the spectra challenging to interpret and limiting the utility of such a labelling scheme. Other tools such as constant-time $^{13}$C-$^{1}$H HSQC spectra[17,18] also do not provide high-quality spectra of large uniformly labelled proteins, since the constant-time substantially skews the intensities and even renders many signals invisible. However, due to the inherent sensitivity of methyl groups, $^{13}$C-$^{1}$H correlation maps of protonated large proteins nonetheless contain a significant amount of information from many of the methyl groups present. The challenge is

that these spectra are difficult to interpret, even by specialists, due to the poor resolution, see e.g. Fig. 1b.

### Training and assessment using synthetic data

In order to transform $^{13}$C-$^{1}$H correlation maps from universally $^{13}$C labelled proteins into spectra that can easily be interpreted, we train two DNNs, both of which are based on the FID-Net architecture[7]. Briefly, the first network is trained to transform time-domain FIDs in the $^{13}$C dimension by removing a single cosine modulation corresponding to a $^{13}$C-$^{13}$C coupling constant and reducing the decay rate of the peak such that it gives a sharper signal in frequency domain. The second DNN is trained to act on FIDs in the $^{1}$H dimension. In this case, the network is trained only to reduce the decay rate of FIDs so that peaks are sharper in the frequency domain of this dimension. Both networks are trained independently solely on synthetic data (full parameters provided in the supplementary information).

For transforming a full $^{13}$C-$^{1}$H 2D plane of a uniformly labelled protein the workflow is as follows: the input 2D plane is first processed and Fourier transformed in the $^{1}$H dimension before being transposed. This half-processed spectrum is then passed to the first DNN where the signal modulation due to one-bond $^{13}$C-$^{13}$C couplings is removed and the signals are sharpened. Subsequently, the $^{13}$C dimension is processed and Fourier transformed as normal. The spectrum is then transposed back to the $^{1}$H dimension, inverse Fourier transformed, and Hilbert transformed. The resulting time-domain data is passed to the second DNN to sharpen signals in the $^{1}$H dimension. This $^{1}$H dimension is then reprocessed, and Fourier transformed to yield the final frequency-domain spectrum.

In order to test and benchmark this approach, it was first applied to synthetic data. Rather than using randomly generated data, as is done in the training of the networks, we aimed to benchmark performance on synthetic data that were nonetheless reminiscent of actual $^{13}$C-$^{1}$H correlation maps of proteins. In order to do this, synthetic spectra were made using chemical shifts expected for real systems as sampled from the BMRB[19]. Using this approach, we generate one hundred synthetic spectra with a comparable number of peaks to the 42 kDa protein HDAC8[20], and one hundred synthetic spectra with a comparable number of peaks to MSG[21] (see ref. 22 for data availability). These spectra contain all expected $^{13}$C-$^{13}$C couplings as well as broad peaks as expected for large, protonated proteins. Given that these spectra are synthetically generated, we can also generate an idealised target spectrum in which all $^{13}$C-$^{13}$C scalar couplings are removed, the linewidths are narrowed, and the positions of all peaks within this target spectrum are known. In order to test the performance of the DNNs, we transform the original synthetic spectra using our two trained DNNs. We then pick peaks in the resulting transformed spectra and compare the results against the known peak positions by quantifying the rate of true positives, false positives, and false negatives. In order to avoid any influence of the accuracy of the peak-picking algorithm used, which can vary, we only considered peaks that are isolated in the processed spectra (with a distance larger than processed linewidths), and those that are doublets in the original spectrum.

As shown in Fig. 2, for synthetic spectra, the FID-Net processing appears successful at significantly enhancing the resolution of spectra expected for uniformly $^{13}$C-labelled proteins, even when there are a number of heavily overlapping signals. Based on the high levels of true positive peaks and low levels of false positive peaks observed in the synthetic data, we proceeded to test the FID-Net based processing on experimental data, where the proteins are uniformly $^{13}$C labelled in the absence of deuteration.

### Application to experimental 2D $^{13}$C-$^{1}$H correlation spectra

To test the feasibility of our approach to transform experimental $^{13}$C-$^{1}$H correlation spectra of uniformly $^{13}$C labelled proteins, the method was

applied to increasingly larger proteins, demonstrating the ability of the FID-Net approach to provide high quality correlation maps, similar in quality to methyl-TROSY spectra. In addition to higher expression yields, a key advantage of the methodology here is that information is provided on all methyl-bearing side chains, including, alanine $^{13}C^{\beta}$, isoleucine $^{13}C^{\gamma2}$ and $^{13}C^{\delta1}$, leucine $^{13}C^{\delta1,\delta2}$, methionine $^{13}C^{\epsilon}$, threonine

$^{13}C^{\gamma2}$, and valine $^{13}C^{\gamma1,\gamma2}$. While methods do exist for specific labelling of nearly all methyl groups[23], these approaches typically lead to reduced yields and come with higher costs.

Following benchmarking on synthetic data, the trained FID-Net networks were first applied to the 42 kDa protein HDAC8[20]. While a protein of this size is relatively small for methyl-TROSY studies, as

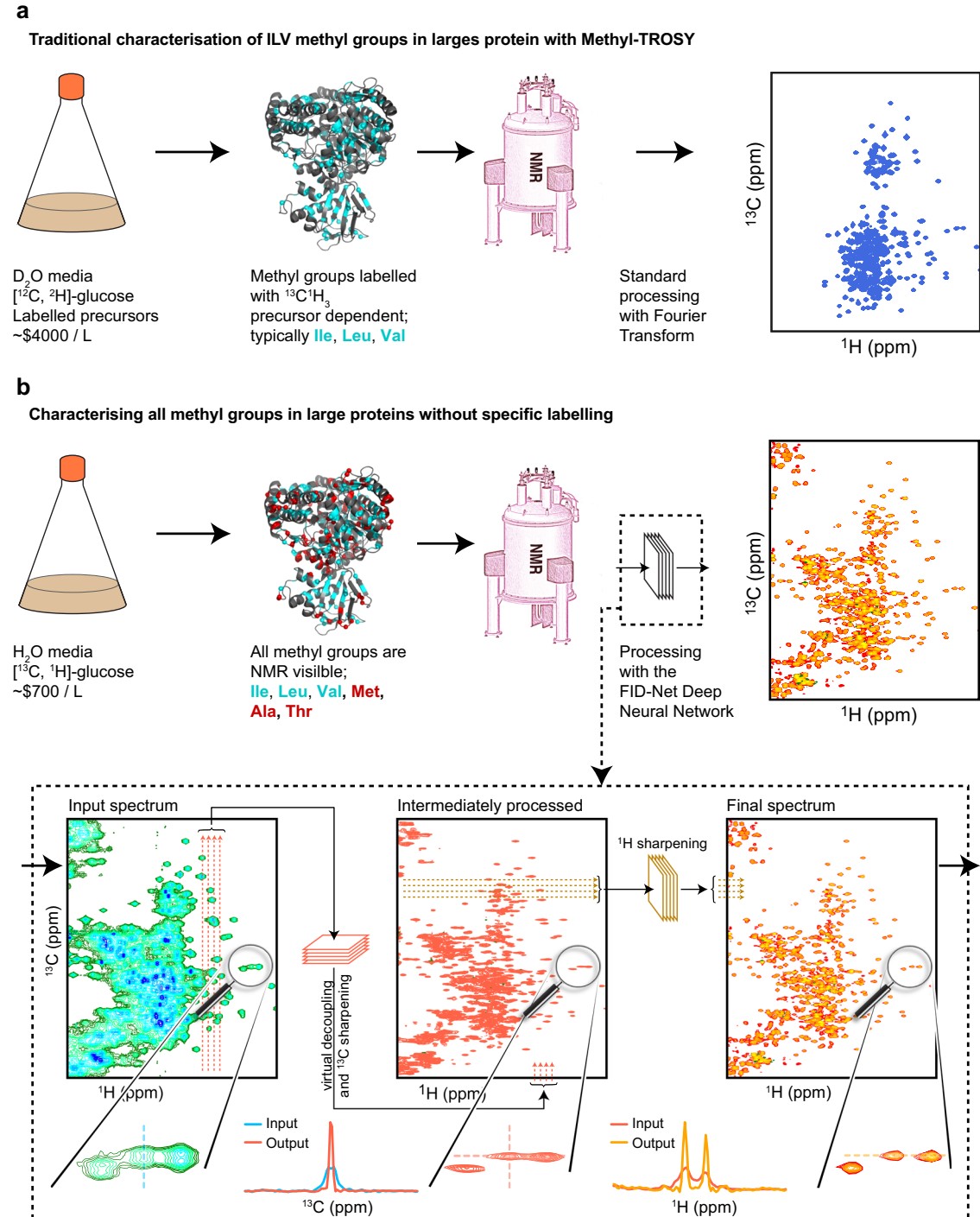

**a**

**Traditional characterisation of ILV methyl groups in larges protein with Methyl-TROSY**

D₂O media
[$^{12}$C, $^{2}$H]-glucose
Labelled precursors
~$4000 / L

Methyl groups labelled with $^{13}C^{1}H_3$ precursor dependent; typically Ile, Leu, Val

Standard processing with Fourier Transform

$^{13}$C (ppm)
$^{1}$H (ppm)

**b**

**Characterising all methyl groups in large proteins without specific labelling**

H₂O media
[$^{12}$C, $^{1}$H]-glucose
~$700 / L

All methyl groups are NMR visilble;
Ile, Leu, Val, Met, Ala, Thr

Processing with the FID-Net Deep Neural Network

$^{13}$C (ppm)
$^{1}$H (ppm)

Input spectrum

$^{13}$C (ppm)
$^{1}$H (ppm)

virtual decoupling and $^{13}$C sharpening

Intermediately processed

$^{13}$C (ppm)
$^{1}$H (ppm)

$^{1}$H sharpening

Final spectrum

$^{13}$C (ppm)
$^{1}$H (ppm)

— Input
— Output

$^{13}$C (ppm)

— Input
— Output

$^{1}$H (ppm)

**Fig. 1 | Overview of processing NMR spectra with FID-Net. a** Overview of traditional tools used to characterise methyl groups in large proteins, which requires expression in bacterial cells such as *E. coli*, deuteration of the protein, and specific isotopic labelling. **b** Overview of our method to characterise large, non-deuterated, uniformly labelled proteins, enabled by the trained deep neural network FID-Net. Two FID-Net networks are trained: (i) one to virtually decouple and enhance the resolution in the $^{13}$C dimension of the initial 2D $^{13}$C-$^{1}$H correlation spectra (green-blue to red spectra), (ii) followed by a second network trained to enhance the resolution in the $^{1}$H dimension (red to orange spectra). The example shown is that of Malate Synthase G (MSG) an 81 kDa protein. As the protein is uniformly labelled it gives rise to peaks associated with all methyl groups in the protein, including methionine, alanine, and threonine residues, as well as isoleucine γ2 methyl groups. The additional methyl probes offered by the uniform labelling scheme are highlighted on the structure of MSG (red). The estimated costs in a and b are calculated using listed prices from Sigma-Aldrich.

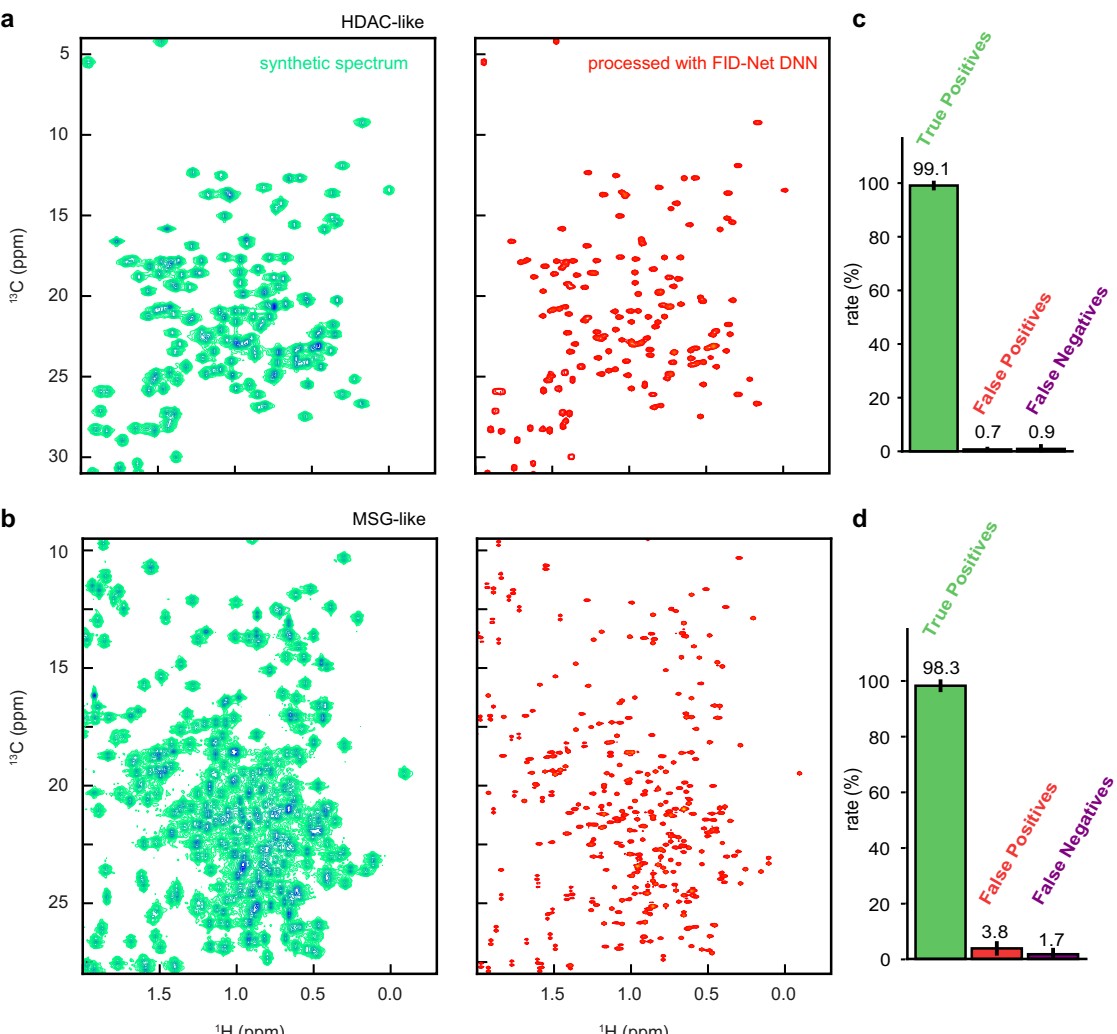

**Fig. 2 | Assessment using synthetic data. a, b** Exemplar synthetic data without processing by the FID-Net DNNs (left column) and with FID-Net based processing (right column). **a** A synthetic spectrum where we have a similar number of signals to HDAC8 (42 kDa). **b** A spectrum that has a similar number of signals to MSG (81 kDa). One hundred distinct spectra with a similar number of signals to those shown in **a** or **b** are generated. These are then analysed using the FID-Net approach and the resulting spectra are peak picked. From FID-Net analysed spectra peaks are picked and compared to ground truth values. From picked peaks, true positive, false positive and false negative rates of peaks are calculated (only considering isolated peaks) and plotted (**c, d**). Full details are given in the Methods section.

shown in the $^{13}$C-$^1$H correlation spectrum in Fig. 3a (blue-green) of a non-deuterated, uniformly $^{13}$C labelled sample, signals in the methyl region of the spectrum are nonetheless broad and overlapped, making many of them difficult to discern. This also holds for a constant-time $^{13}$C-$^1$H HSQC spectrum, where the constant-time (27 ms or 54 ms) substantially skews intensities and renders many of the signals invisible, see Supplementary Fig. 5. Conversely, following application of the FID-Net Networks (orange spectrum), Fig. 3b, the signals are virtually decoupled in the $^{13}$C dimension and sharpened in both the $^{13}$C and $^1$H dimensions. This dramatically simplifies peak identification. By overlaying the FID-Net transformed spectrum with a classical methyl-TROSY spectrum of an ILVM specifically labelled sample of HDAC8 (blue), Fig. 3c, where only the side chains of these amino acids are labelled, an excellent correspondence is seen between isoleucine, leucine, and valine methyl peaks. The linewidths of the peaks in both of these spectra are highly comparable and all expected peaks from the methyl-TROSY spectrum are recovered in the FID-Net processed spectrum of the uniformly labelled sample. Additional peaks are also visible in the FID-Net processed spectrum, due to the presence of additional labelled methyl groups, including, threonine and isoleucine $^{13}$C$^{γ2}$. Small peak-shifts are mainly due to the isotope shifts originating

from deuteration[24]. A full overlay of the FID-Net processed spectrum of HDAC8 and the methyl-TROSY spectrum is shown in Supplementary Fig. 1. All previously assigned methyl cross-peaks for HDAC8[25] were present in the FID-Net processed spectra of HDAC8.

To test the robustness of the FID-Net processing approach on larger systems, with substantially more cross-peaks and signal overlap, we next applied the FID-Net DNNs to study the methyl region of the protein MSG. This 723-residue protein has been studied extensively by NMR[21,26], but all of these studies have required deuteration to minimise the broadening of signals due to extensive relaxation. However, as shown in Fig. 3e, by coupling the intrinsic sensitivity of methyl groups with FID-Net processing, it is possible to obtain high-quality methyl-TROSY like spectra for this system at a lower cost and with the added bonus of signals associated with all methyl bearing side chains. As with the HDAC8 example above, clear agreement between the expected peaks in the ILV spectrum and FID-Net processed spectrum is attained and the linewidths in these two spectra are also similar. A full overlay of the FID-Net processed spectrum of MSG and the methyl-TROSY spectrum is shown in Supplementary Fig. 2. The FID-Net processed spectrum of MSG also agrees with the previously published chemical shift assignments for $^{13}$C$^{γ2}$ methyl groups of Ile and Thr, $^{13}$C$^ε$ of Met, and

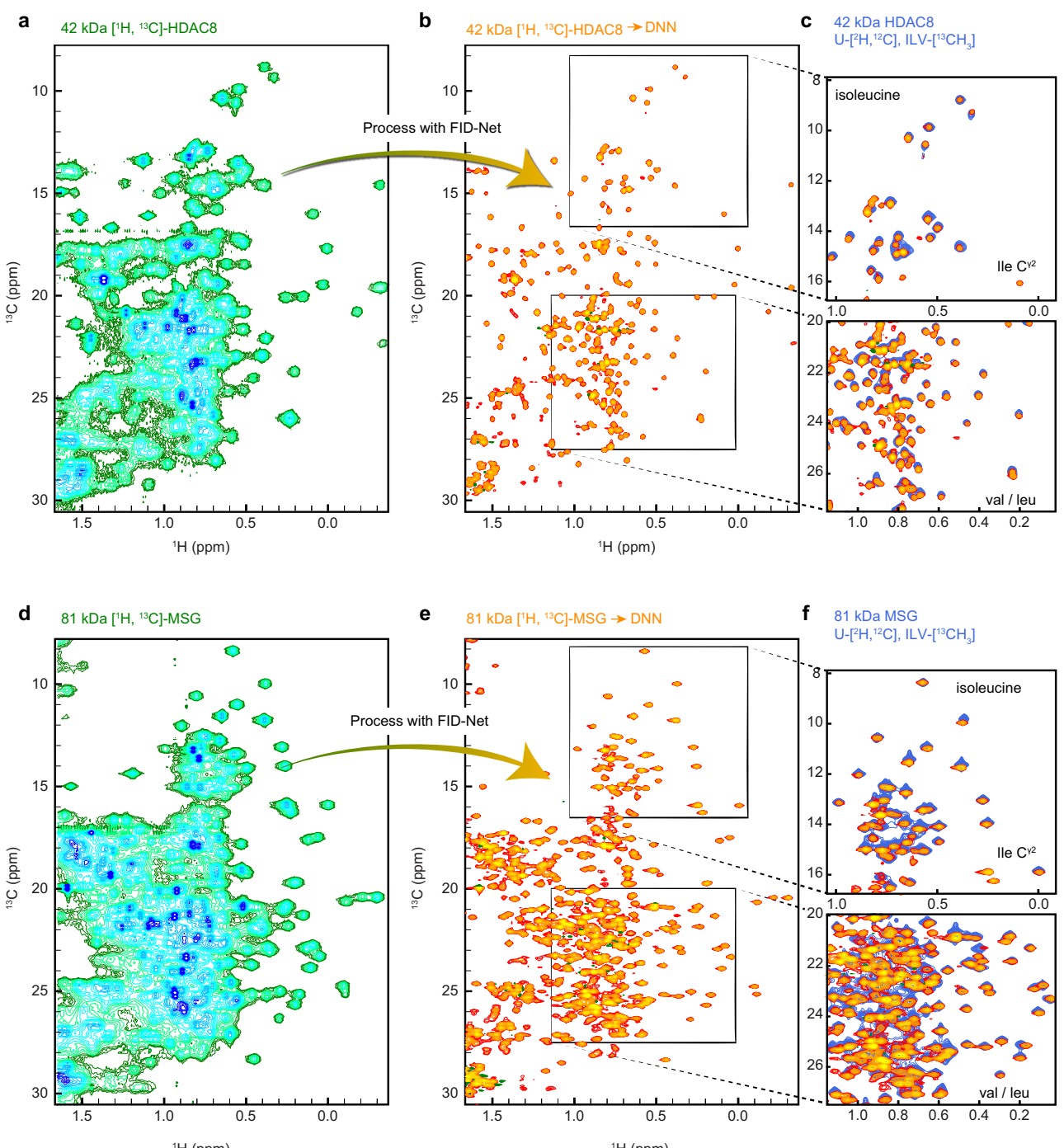

**Fig. 3 | FID-Net processed methyl HSQC spectra of uniformly $^{13}$C labelled non-deuterated proteins. a** A $^{13}$C-$^1$H HSQC NMR spectrum of uniformly $^{13}$C labelled, non-deuterated, HDAC8 (42 kDa) processed with a standard discrete Fourier transform. **b** The spectrum in **a** processed with the FID-Net DNNs. **c** Comparison of the FID-Net processed HSQC spectrum in **b** (orange) with a methyl-TROSY HMQC spectrum of ILV specifically labelled and deuterated HDAC8 (blue). **d** A $^{13}$C-$^1$H HSQC NMR spectrum of uniformly $^{13}$C labelled MSG processed with a standard discrete Fourier transform. **e** The spectrum in **d** processed with the FID-Net DNNs. **f** Comparison of the FID-Net processed HSQC spectrum in **d** (orange) with a methyl-TROSY HMQC spectrum of ILV specifically labelled and deuterated MSG (blue) for two selected regions. As expected, many methyl groups are not visible in the ILV labelled sample, such as, Isoleucine $^{13}$C$^{\gamma 2}$ (labelled).

$^{13}$C$^\beta$ of Ala[27–29], with the exception of $^{13}$C$^\beta$ of A633, which was not observed in the FID-Net processed spectra.

To push the limits of the proposed method we tested its performance on the 360 kDa α7α7 (half-proteasome) from *T. acidophilum*. This protein complex has an effective rotational correlation time of approximately 120 ns at 50 °C[30]. The high degree of symmetry in the complex (composed of 14 monomeric units that form two heptameric rings) means that there are relatively few peaks in its spectra compared

to its size, see Fig. 4. A full overlay of the FID-Net processed spectrum of α7α7 and the methyl-TROSY spectrum is shown in Supplementary Fig. 3. In the case of the 360 kDa α7α7 complex, it is clear that a number of peaks present in the deuterated, ILV labelled sample are fairly weak in the FID-Net decoupled spectrum (marked with an asterisk in Fig. 4), this is particularly evident in the isoleucine δ$_1$ region of the spectrum, and we therefore judge that currently systems such as the α7α7 complex are at the limit of our approach. Additional peaks in the

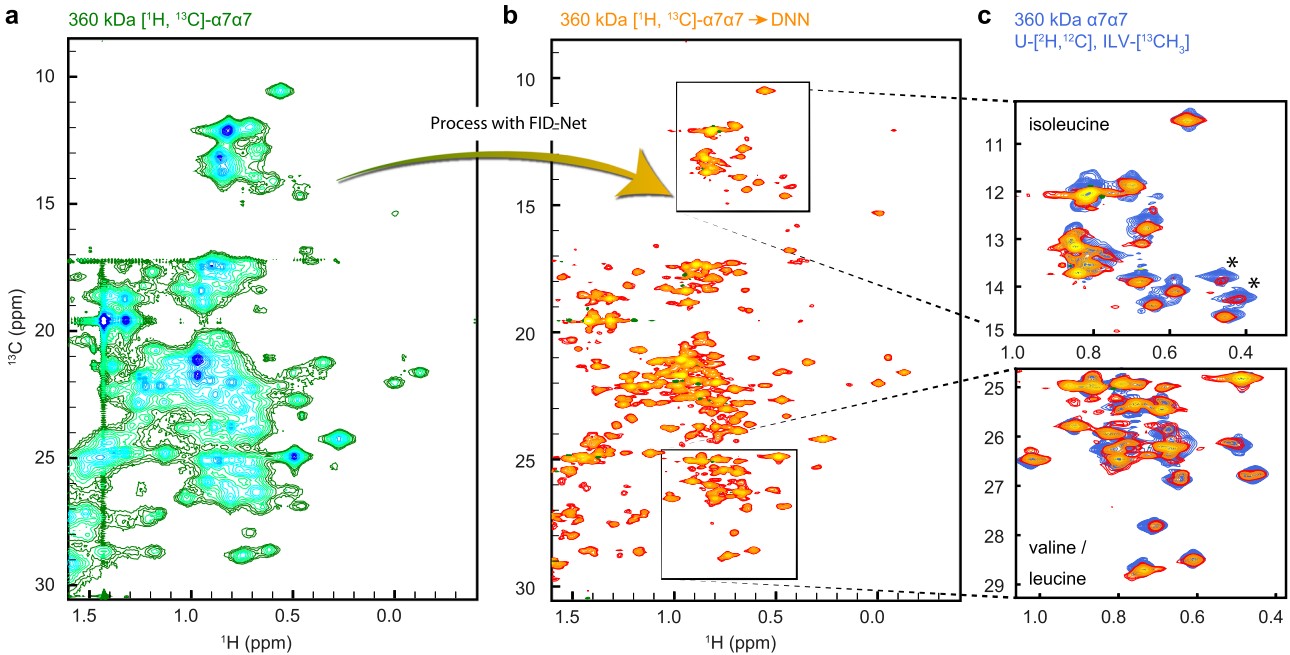

**Fig. 4 | Application to a 360 kDa complex. a** A $^{13}C$-$^1H$ HSQC NMR spectrum of uniformly $^{13}C$ labelled, non-deuterated, α7α7 (360 kDa) processed with a standard discrete Fourier transform. **b** The spectrum in **a** processed with the FID-Net DNNs. **c** Comparison of FID-Net processed HSQC spectrum in **b** (orange) with a methyl-TROSY HMQC spectrum of an ILV specifically labelled and deuterated α7α7 (blue).

spectrum are also clearly visible due to the presence of alanine, threonine, methionine and isoleucine $^{13}C^{\gamma 2}$ methyl resonances.

While for the proteins shown above the isoleucine $^{13}C^{\delta 1}$, leucine $^{13}C^{\delta 1,\delta 2}$ and valine $^{13}C^{\gamma 1,\gamma 2}$ methyl group resonances can be readily compared to those obtained in ILV-labelled samples using traditional methyl-TROSY spectra, peaks associated with alanine, threonine, methionine, and isoleucine $^{13}C^{\gamma 2}$ methyl groups are less readily available. To verify the reliability of these additional peaks observed in uniformly $^{13}C$-labelled samples and demonstrate the extension of the methodology to 3D spectra, we recorded a $^{13}C$-$^{13}C$-$^1H$ NOESY spectra of MSG and, as demonstrated below, use this methodology to provide assignments for threonine, methionine, and isoleucine side chains.

**Applications to three-dimensional NOESY spectra**
Following the successful implementation of deep neural networks for the production of methyl spectra of similar quality to typical deuterated methyl-TROSY spectra just from uniformly $^{13}C$ labelled samples in a protonated background, we applied the methodology to a 3D $^{13}C$-HSQC-NOESY-HSQC experiment acquired on a uniformly $^{13}C$ labelled sample of MSG made in (non-deuterated) water. Despite the high molecular weight of MSG, we observed NOE cross-peaks among the inter methyl protons that were within a distance of 3.0 Å to 5.0 Å of each other (Figs. 5a–d and Supplementary Fig. 4).

In total, 292 NOE cross-peaks were observed among 170 methyl-bearing residues from different regions of the protein. Furthermore, like conventional NOESY spectra, we observed a correlation between the NOE cross-peaks volumes ($V$) and the distance between the proton pairs ($r$), i.e. $V \propto 1/r^6$, Fig. 5f. Using this experiment, we could easily characterise the contact between two methyls, for example, $^{13}C^{\delta 1}$ and $^{13}C^{\gamma 2}$ of isoleucines as shown in Supplementary Fig. 4a. Similarly, the two geminal methyl resonances of leucine and valine can be linked using this spectrum (Fig. 5d). However, a combination of 3D $^{13}C$-HSQC-NOESY-HSQC and 3D HMBC-HMQC experiments would be the best method to link the geminal methyl resonances of leucine and valine without any ambiguity[31]. Additionally, using this approach, NOEs can be observed between methyl protons of all methyl-bearing residues (isoleucine, leucine, valine, methionine, alanine, and threonine) using

one sample, which is not possible in the conventional method due to metabolic scrambling of amino acids in selective $^{13}C$ labelling[32]. Therefore, the DNNs can be utilised to produce a 3D $^{13}C$-HSQC-NOESY-HSQC experiment acquired on a uniformly $^{13}C$ labelled sample without deuteration, resulting in a spectrum of similar quality to a 3D $^{13}C$-HMQC-NOESY-HMQC experiment acquired on a specifically methyl labelled sample. This approach can be generally applied to large proteins and complexes without deuteration.

## Discussion
The ability to characterise the regulation, interactions, and dynamics of large proteins in solution is paramount to understanding their molecular functions. The methyl-TROSY methodology is one of the most important developments in biomolecular NMR over the last decades and these methods have truly paved the way for NMR to offer key insights on larger biomolecular complexes, complementing other structural approaches including cryo-electron microscopy and AI-based predictions[33]. However, the labelling requirements for such NMR experiments are demanding, ideally requiring perdeuteration and the use of specific precursors to introduce [$^1H$,$^{13}C$]-labelled methyl moieties of specific locations. While the resulting spectra are of high-quality, the cost of such labelling is higher, typically leads to lower protein yields, and is inconsistent with protein production methods for many systems of interest such as eukaryotic and membrane proteins. Above, we presented an alternative method to classical methyl-TROSY NMR for characterising large proteins in solution, which is based on uniformly protonated, $^{13}C$-labelled samples and processing with FID-Net neural networks. With the approach one can characterise proteins up to about 350 kDa. The size limitation for FID-Net based processing arises because it is reliant on some signal still being present in the spectra and above this size most signals, even those associated with methyl moieties, are broadened beyond detection in non-deuterated systems.

The main disadvantage of the FID-Net method is that the process of peak-sharpening inevitably leads to a loss of the intrinsic shape of the cross-peaks, including the peak height. Accurately measuring peak intensities is critical in a number of NMR experiments, including

diffusion and relaxation, so it is not advised to record these experiments in conjunction with the presented FID-Net processing nor is it advised to perform line-shape analyses. However, for a large body of NMR experiments, the main parameter of importance is the chemical

shift as well as a reasonable estimate of the peak intensity, and in these cases we believe that FID-Net processing will prove useful. Such studies include classical chemical shift perturbation studies, such as, changes in chemical shifts upon addition of an interacting partner or changes

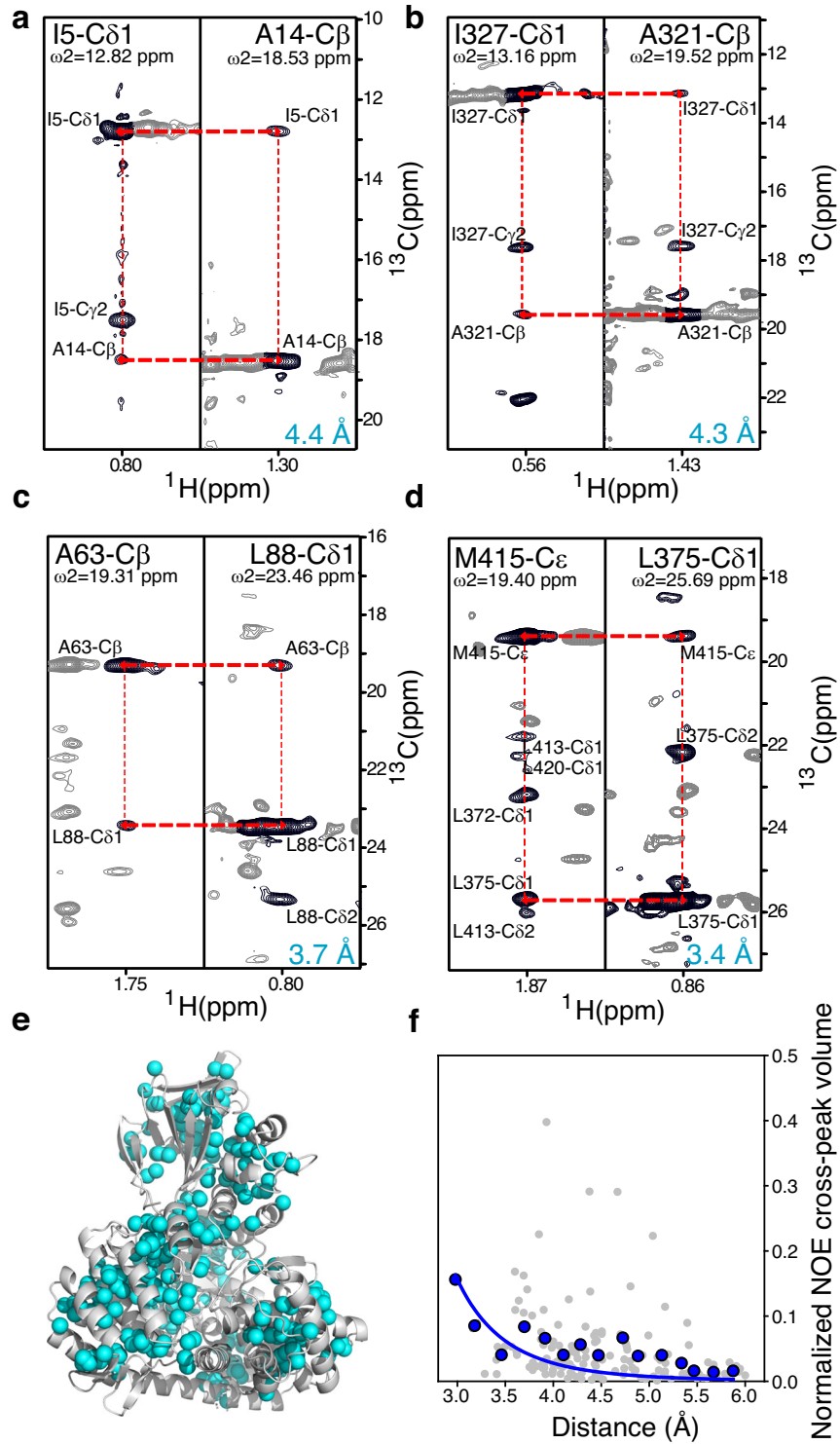

**Fig. 5 | NOESY spectra of non-deuterated 80 kDa MSG.** Two-dimensional planes from the 3D $^{13}$C-$^{13}$C-$^{1}$H NOESY spectrum for **a** I5-$^{13}$C$^{δ1}$ and A14-$^{13}$C$^{β}$. **b** I327-$^{13}$C$^{δ1}$ and A321-$^{13}$C$^{β}$. **c** A63-$^{13}$C$^{β}$ and L88-$^{13}$C$^{δ1}$. **d** M415-$^{13}$C$^{ε}$ and L375-$^{13}$C$^{δ1}$. **e** Methyl groups of isoleucine, leucine, valine, methionine, alanine, and threonine showing NOE cross-peaks in the 3D $^{13}$C-$^{13}$C-$^{1}$H NOESY spectra are highlighted as cyan spheres on a cartoon representation of Malate Synthase G (MSG) structure (PDB entry:1D8C). **f** Normalised NOE cross-peak volumes (cross-peak volume/diagonal-peak volume)

*versus* interproton distances. Grey circles represent the normalised NOE cross-peak volumes obtained for individual NOE cross-peaks, whereas blue circles represent the average of normalised NOE cross-peaks volume over interproton-distance intervals of 0.2 Å, i.e. (Sum of normalised NOE cross-peak volumes) / (Number of cross-peaks), within each interproton-distance interval of 0.2 Å. The blue line represents the fitted curve of NOE cross-peaks volume ($V$) and interproton distance ($r$) using the standard equation $V = C/r^6$, where $C$ is a constant.

related to varying conditions, *e.g.* pH. As we have shown above, the obtained intensities are of a sufficient quality such that the transformed spectra facilitate facile chemical shift assignment of methyl peaks by either NOESY spectra (as demonstrated here for MSG) or by point mutations, which often requires several samples. We are currently in the process of developing and training a DNN that is able to enhance the resolution of $^{13}C$-$^1H$ spectra and provide quantitative mappings, which can be used for downstream analyses, *e.g.* relaxation experiments. This work remains ongoing.

A number of approaches have previously been suggested to overcome the limitations of methyl-TROSY highlighted above, particularly when making perdeuterated samples is not possible. Recent examples include the use of delayed decoupling, as has also been used for very large complexes with molecular weights in the MDa range[34], optimised NMR pulse sequences to probe methionine residues in proteins with molecular weights up to 240 kDa[32], and the use of local deuteration of leucine residues to probe their methyl groups in membrane and insect cell derived proteins[35]. While very powerful, these methods are limited in that they only consider a single residue type, thus restricting the number of available probes in the system. Conversely, the methodology developed here offers simultaneous access to all methyl bearing side chains in a protein, offering many more probes of biomolecular behaviour. By decoupling signals in the $^{13}C$ dimension and sharpening them in both the $^1H$ and $^{13}C$ dimensions, the resulting spectra resemble those given by perdeuterated samples with specific methyl labelling.

We believe that the FID-Net methodology presented here will significantly lower the barrier to entry for NMR of large systems. While methods exist for obtaining methionine and threonine assignments for well-studied systems such as MSG, processing with the FID-Net DNNs provides a straightforward approach, which does not require perdeuteration and offers insight into all methyl-bearing residues. We envisage that the idea of using DNNs for peak sharpening and simultaneous homonuclear virtual decoupling within NMR could be applied in other cases to improve spectra and that processing NMR data with DNNs will facilitate additional ventures within NMR. As such we see the presented method as paving the way for a plethora of ways for generally analysing and transforming NMR spectra with deep neural networks to push the capabilities and limits of NMR.

## Methods

### Initial considerations about the neural networks

In the present study our aim was to develop DNNs to map $^{13}C$-$^1H$ methyl correlation spectra of large uniformly $^{13}C$-labelled proteins into spectra that are similar to methyl-TROSY spectra of highly deuterated proteins. Two objectives must be fulfilled to achieve this aim: (*i*) the one-bond $^{13}C$-$^{13}C$ homonuclear scalar couplings associated with methyl groups must be decoupled and (*ii*) the peaks must be sharpened, making them more easily resolvable, equivalent to slowing down the exponential decay of magnetisation in the time domain. It should be noted that these changes do not increase the information content in the spectrum, but they do make the information contained within the original spectra more easily interpretable by spectroscopists.

We employ the FID-Net architecture that we have previously shown to successfully perform a number transformations on time domain NMR data, including reconstructing non-uniformly sampled spectra[7,9,14] and homonuclear virtual decoupling[8]. The FID-Net architecture currently only transforms a set of 1D spectra, and two separate FID-Net DNNs were therefore trained: one was optimised for spectral parameters typically encountered in the $^{13}C$ dimension and trained to both decouple and sharpen signals, whereas the second FID-Net DNN was optimised for spectral parameters in the $^1H$ dimension alone and was trained to only sharpen signals. A schematic illustration and summary of the effects of the neural networks is provided in Fig. 1b. In both cases the networks are trained and validated exclusively on synthetic data and then tested on experimentally acquired data.

Care must be taken when determining how and to what extent signals should be sharpened using DNNs. For example, signals from flexible regions of proteins that already give rise to sharp peaks could result in the presence of truncation artefacts in the spectrum. On the other hand, broad signals require significant attenuation of their relaxation to be clearly resolvable in the frequency domain. The broader the signal, the more attenuation of relaxation that is therefore desirable. To satisfy these requirements the following function is used for input $R_2^{in}$ and target $R_2^{tar}$ transverse relaxation rates in the training data:

$$R_2^{tar} = \max\left( R_2^{max}\tanh\left(\frac{R_2^{in}}{R_2^{max}}\right), R_2^{max}\left(1 - \tanh\left(\frac{R_2^{in}}{R_2^{max}}\right)\right)\right) \quad (1)$$

The effect of this is to make the linewidths in the target spectrum relatively uniform similar to methyl-TROSY spectra, where for the DNN relaxation rates above $R_2^{max}$ are scaled down towards it, while those below $R_2^{max}$ are scaled up towards it. A value of $R_2^{max} = 25\,s^{-1}$ was chosen so that target spectra have linewidths similar to those observed in relatively structured residues in an ILV sample of a medium-to-large deuterated protein. Further details of the neural network architecture, training data parameters, and training procedures are provided in the supplementary information.

Once trained, the neural networks can easily be applied as part of processing scripts, examples of which are provided in the supplementary information. The DNNs are trained on a diverse range of NMR parameters (Supplementary Tables 1 and 2) and thus can be used without further retraining. The approach can be used with standard $^1H$-$^{13}C$ HSQC or HMQC pulse sequences (vide infra).

### The network architecture and training

Two networks were trained for the study: one for removing $^{13}C$-$^{13}C$ couplings and sharpening spectra in the $^{13}C$ dimension and the second solely for sharpening spectra in the $^1H$ dimension (as described above). Both networks used the previously described FID-Net architecture. The input size for the $^{13}C$ network is $1024 \times 4$ and for the $^1H$ network is $512 \times 4$. As was the case with previous FID-Net architectures, the networks consist of a series of stacked residual units, wherein each residual unit consists of dilated convolutional layers with kernel size $8 \times 4$. The filters are activated by either sigmoidal (50%) or tangent (50%) functions. The results of the activations are then multiplied and passed through another convolutional layer with kernel size $8 \times 4$. The output from each layer is combined to give the final output and also added to the input for the residual unit to form the input for the next layer. For the $^{13}C$ network, the dilations employed are cycled through the values: 1, 2, 4, 6, 8, 10, 12, 14, 16, 20, 24, 28, 32, 40, 48, 56, 64, and there are 128 filters for each convolutional layer. For the $^1H$ network the dilations employed are 1, 2, 4, 6, 8, 10, 12, 14, 16, 20, 24, 28, 32, and there are 64 filters per convolutional layer.

For each network 500,000 test planes were created for training and 50,000 for testing using the parameters given in Supplementary Tables 1 and 2. The models were developed and trained using the Tensorflow library[36] with the Keras-front end[37]. The cost function used to train the networks is the mean squared error in the frequency domain between the spectrum produced by the DNN and the target spectrum wherein the linewidth of peaks are set according to the $R_2$ scaling described above and for the $^{13}C$ network the scalar coupling is removed. The RMSprop optimizer[38] was used in training. For both networks the learning rate was initially set to $10^{-4}$ until the validation loss value plateaued and was then reduced to $10^{-5}$ until it plateaued again where training was then ended.

**Table 1 | Parameters used for benchmarking[a]**

| Parameter | HDAC-like spectra | MSG-like spectra |
|---|---|---|
| Number of signals | 275 | 600 |
| Larmor Frequency (MHz) | $\in (600,700,800,950)$ | $\in (600,700,800,950)$ |
| $^1$H SW (Hz) | 2000–5000 | 2000–5000 |
| $^{13}$C SW (Hz) | 2000–000 | 2000–5000 |
| $^1J_{CC}$ (Hz) | 34 (2) | 34 (2) |
| $R_2^{(1)}$ (s$^{-1}$) ($^1$H dim) | 50 (10) | 60 (15) |
| $R_2^{(2)}$(s$^{-1}$) ($^{13}$C dim) | 50 (10) | 60 (10) |

[a] Where values are given as a range, for each spectrum the true value is taken as a random value from a uniform distribution of the range. Where values are given as a number followed by a bracketed number, for each signal the value used is randomly chosen from a normal distribution centred on the first number and with a standard deviation given by the bracketed value. For all synthetic spectra, half of the peaks are chosen as resulting from a methyl moiety, while the other half come from non-methyl $^{13}$C-$^1$H moieties, i.e., leading to triplets in the $^{13}$C dimension.

## Benchmarking using synthetic data

Once trained, to validate the performance of the two networks we use synthetically generated spectra. Rather than using arbitrary spectra as was done for training the networks, we attempt to generate realistic $^{13}$C-$^1$H correlation maps for uniformly labelled $^{13}$C proteins using chemical shift statistics from the BMRB[19]. In addition to containing terminal $^{13}$C moieties that give rise to doublets in the spectra as a result of a single $^{13}$C-$^{13}$C scalar coupling these spectra also contain multiplets due to moieties that have multiple $^{13}$C-$^{13}$C scalar couplings (though these are usually at higher $^1$H frequencies as is observed in real spectra).

We generate 200 synthetic spectra in total: the first 100 spectra were chosen to have features similar to a $^{13}$C-$^1$H spectra of a protein with a similar size to HDAC8 whilst the second 100 were chosen to have features similar to a protein with a similar size to MSG. Example spectra are shown in Figs. 2a and 2b. The parameters used to generate the synthetic spectra for benchmarking are detailed in Table 1.

Once the synthetic spectra are made, they are processed using the deep neural network pipeline. Visual inspection of the transformed spectra suggests that the method is effective at decoupling and sharpening spectra, such that they can be interpreted more easily. We also note that where multiple $^{13}$C-$^{13}$C couplings are present, the network will remove just one of the couplings such that triplets become doublets. To evaluate the performance of the pipeline quantitatively, the transformed spectra are peak picked using the built-in peak picker in NMRPIPE. The results are compared to the known ground-truth values for peak positions (assuming no couplings were present in the spectra).

We focus on two key parameters in this evaluation: the number of true positive picked peaks and the number of false positive picked peaks. Given the difficulty of picking peaks from crowded regions automatically, and that our aim is to evaluate the performance of our alternative pipeline for analysing spectra we focus on isolated peaks where the performance of the peak picker is robust. Here, we define an isolated peak as any ground truth peak where the minimum distance from any other ground truth peak is greater than or equal to 0.06 $^1$H ppm (0.24 $^{13}$C ppm). Furthermore, we ignore peaks that are part of doublets and that are more than 1.50 $^1$H ppm from a target peak originating from a methyl moiety. Once peaks in the FID-Net processed spectrum have been picked, they are matched with the closest peak in the target peak list. Each peak can only be matched with a single target peak and we match in descending order of distance between picked peaks and target peaks until there are no picked peaks within the minimum distance to a target peak (set at 0.03 $^1$H ppm) or there are no picked nor target peaks left. This process is then repeated for all synthetic spectra.

The true positive rate is defined as the percentage of peaks where we see a clear correspondence between a picked and target peak. The false positive rate is defined as the rate at which a peak identified in the FID-Net processed spectrum does not correspond with any peaks in the actual spectrum. Given the limitations of peak pickers, the true and false positive rates here likely represent a lower bound on the performance of the FID-Net processing pipeline.

## General isotopic labelling

In this study, two different types of labelled protein samples were used for acquisition of the NMR data: (1) uniformly $^{13}$C,$^{15}$N labelled sample made in $^1$H$_2$O. (2) methyl Ile-$^{13}$C$^{δ1}$,$^1$H$^{δ1}$, Leu-$^{13}$C$^{δ1}$, $^1$H$^{δ1}$/$^{13}$C$^{δ2}$,$^1$H$^{δ2}$, and Val-$^{13}$C$^{γ1}$,$^1$H$^{γ1}$/$^{13}$C$^{γ2}$,$^1$H$^{γ2}$ labelled sample made in deuterated background. To express uniformly $^{13}$C,$^{15}$N labelled proteins, we used M9 media that was made with $^1$H$_2$O and supplemented with 1 g/L [$^1$H,$^{15}$N]-ammonium chloride and 3 g/L of [$^1$H,$^{13}$C]-glucose as the sole nitrogen and carbon sources. For expression of methyl Ile-$^{13}$C$^{δ1}$,$^1$H$^{δ1}$, Leu-$^{13}$C$^{δ1}$, $^1$H$^{δ1}$/$^{13}$C$^{δ2}$,$^1$H$^{δ2}$, and Val-$^{13}$C$^{γ1}$,$^1$H$^{γ1}$/$^{13}$C$^{γ2}$,$^1$H$^{γ2}$ labelled proteins, we used $^2$H$_2$O M9 media supplemented with 1 g/L [$^1$H,$^{15}$N]-ammonium chloride and 3 g/L of [$^2$H,$^{12}$C]-glucose as the sole nitrogen and carbon sources. Methyl labelling was achieved by the addition of 60 mg/L alpha-ketobutyric acid [U-$^{12}$C/$^2$H, methyl-$^{13}$CH$_3$] for labelling of isoleucines, and 90 mg/L α-ketoisovaleric acid [U-$^{12}$C/$^2$H, methyl-($^{13}$CH$_3$,$^{12}$CD$_3$)] for labelling of valine and of leucine methyl groups. These precursors were added one hour prior to induction.

## Expression and purification of Histone deacetylase 8 (HDAC8)

The Human HDAC8 construct described by Vannini et al. with a C-terminal 6X-histidine tag in ampicillin-resistant pET21b expression vector was transformed in BL21(λDE3) E. coli cells for protein expression[20,39]. A single colony from the transformed plate was inoculated in 10 ml of LB media supplemented with ampicillin (100 μg/ml) at 37 °C. Once the LB culture reached an OD600 between 0.8 and 1.0, it was used to inoculate a 50 ml M9 minimal media pre-culture. This M9 pre-culture was used to inoculate 1 L of M9 media and grown at 37 °C to OD600 ≈ 0.8. HDAC8 expression was induced for >16 h with 0.5 mM IPTG and 200 μM of ZnCl$_2$ at 21 °C. The cell pellet, collected by centrifugation, was re-suspended in lysis buffer containing 50 mM Tris–HCl pH 8.0, 3 mM MgCl$_2$, 500 mM KCl, 10 mM imidazole, 5% glycerol, and 10 mM β-mercaptoethanol. Later, sonication was performed to lyse the cells after addition of small amounts of DNAse, lysozyme, protease inhibitors tablets (1 tablet per 50 ml, Roche), and 0.25 % IGEPAL. The supernatant fraction of the lysate after centrifugation at 40,000 x g for one hour was purified by Ni-NTA affinity chromatography using a linear imidazole gradient (10–250 mM) in lysis buffer. Further, size-exclusion chromatography using a Superdex-75 column (GE Healthcare) was carried out in buffer containing 50 mM Tris–HCl pH 8.0, 150 mM KCl, 1 mM TCEP, and 5% glycerol. Fractions containing purified HDAC8 was pooled together and concentrated by 10 kDa cut off Amicon (Millipore) ultra-filtration membranes. The concentrated sample was buffer exchanged into NMR-buffer (50 mM K$_2$HPO$_4$ pH 8.0, 30 mM KCl, 4 mM DTT, and 1 mM NaN$_3$) for NMR data acquisition.

## Expression and purification of Malate Synthase G (MSG)

A small adjustment was made to the methods previously described for producing isotopically labelled MSG[26,31,40]. Briefly, to produce the MSG protein, BL21 (DE3) E. coli cells were transformed with a kanamycin-resistant pET28a vector containing MSG gene with a C-terminal 6X-histidine tag. The protein expression protocol for MSG is same as HDAC8 up to induction. MSG expression was induced for >16 h with 1 mM IPTG at 21 °C. The cell pellet was collected by centrifugation and re-suspended in lysis buffer containing 20 mM Tris–HCl pH 7.8, 300 mM NaCl, 10 mM imidazole, and 10 mM β-mercaptoethanol. The protein purification protocol for MSG is the same as for HDAC8, until

the Ni-NTA affinity chromatography. The fractions containing MSG from Ni-NTA affinity chromatography were further purified by size exclusion chromatography using a Superdex-200 column (GE Healthcare) in buffer containing 20 mM Sodium phosphate pH 7.1, and 5 mM dithiothreitol. After gel filtration, the fractions containing pure protein were pooled, concentrated, and buffer exchanged into NMR-buffer (20 mM Sodium phosphate buffer pH 7.1, 5 mM DTT, 20 mM MgCl₂, 1 mM NaN₃) for NMR data acquisition using 30 kDa cut off Amicon (Millipore) ultra-filtration membranes.

### Expression and purification of α-subunit complex (α7α7) of proteasome from *T. acidophilum*

In order to express the α-subunit complex (α7α7) proteasome from *T. acidophilum*, the αWT clone with N-terminal Histidine tag and a TEV protease site was transformed into BL21 (λDE3) *E. coli* cells[31,41]. The protein expression protocol for MSG was followed for the α-subunit complex up to the induction step. The α-subunit complex culture was induced at OD600 ≈ 0.9 with 1 mM IPTG at 37 °C for 5 h. Afterwards, we lysed the cells with sonication in a lysis buffer (50 mM NaH2PO4 pH 8.0, 0.2 M NaCl, 10 mM imidazole) and purified them using Ni-NTA chromatography as described above for the purification of HDAC8 and MSG. After Ni-NTA affinity chromatography, TEV protease was introduced to cleave the 6X-histidine tag before dialyzing the protein against 2 L of dialysis buffer (50 mM Tris-HCl pH 8.0, 1 mM EDTA, 5 mM β-mercaptoethanol) overnight at 4 °C. The TEV cleavage of the protein was followed by another Ni-NTA affinity chromatography to eliminate the histidine tag and un-cleaved protein. Afterwards, size exclusion chromatography was performed using a Superdex 200 column (GE Healthcare) in a buffer containing 50 mM NaH₂PO₄ pH 7.5, and 100 mM NaCl. The fractions containing pure protein were concentrated and buffer exchanged into the NMR buffer (20 mM potassium phosphate pH 6.8, 50 mM NaCl, 1 mM EDTA, 2 mM DTT, 0.03% NaN₃) for NMR data acquisition using 30 kDa cut off Amicon (Millipore) ultra-filtration membranes.

### NMR Acquisition of Two-dimensional ¹³C-¹H correlation spectra

The 2D HSQC spectrum of HDAC8 used as input for FID-Net was recorded on a uniformly [¹³C,¹⁵N]-labelled sample using a standard pulse programme with presaturation on a Bruker 700 MHz Avance III spectrometer equipped with Z-gradient triple-resonance TCI cryoprobe at 298 K and running Topspin 3.6.3. The data was acquired with 1024 and 256 complex points in the ¹H and ¹³C dimensions, respectively, with spectral widths of 10000 Hz and 5000 Hz. 16 scans were obtained per individual FID. 2D Methyl-TROSY spectrum of HDAC8 used for experimental verification was recorded on an ILVM specifically labelled sample on a Bruker 800 MHz Avance III spectrometer equipped with Z-gradient triple-resonance TCI cryoprobe. The data was acquired with 1024 and 256 complex points in the ¹H and ¹³C dimensions, respectively, with spectral widths of 12500 Hz and 4500 Hz. 4 scans were obtained per individual FID.

The 2D HSQC spectrum of MSG used as input for FID-Net was recorded on a uniformly [¹³C,¹⁵N]-labelled sample using a standard pulse programme with presaturation on a Bruker 800 MHz Avance III spectrometer equipped with Z-gradient triple-resonance TCI cryoprobe at 310 K and running Topspin 3.6.3. The data was acquired with 1024 and 256 complex points in the ¹H and ¹³C dimensions, respectively, with spectral widths of 12500 Hz and 5000 Hz. 16 scans were obtained per individual FID. The 2D methyl-TROSY spectrum of MSG used for experimental verification was recorded on an ILV specifically labelled sample on a Bruker 800 MHz Avance III spectrometer equipped with Z-gradient triple-resonance TCI cryoprobe and running Topspin 3.6.3. The data was acquired with 1024 and 192 complex points in the ¹H and ¹³C dimensions, respectively, with spectral widths of 10500 Hz and 5000 Hz. 16 scans were obtained per individual FID.

The 2D HSQC spectrum of α7α7 used as input for FID-Net was recorded on a uniformly [¹³C,¹⁵N]-labelled sample using a standard pulse programme with presaturation on a Bruker 950 MHz Avance HD spectrometer equipped with Z-gradient triple-resonance TCI cryoprobe at 323 K. The data was acquired with 1024 and 256 complex points in the ¹H and ¹³C dimensions, respectively, with spectral widths of 15200 Hz and 5263 Hz. 80 scans were obtained per individual FID. 2D Methyl-TROSY spectrum of α7α7 used for experimental verification was recorded on an ILV specifically labelled sample using a standard pulse programme on a Bruker 800 MHz Avance III spectrometer equipped with Z-gradient triple-resonance TCI cryoprobe at 323 K and running Topspin 3.6.3. The data was acquired with 768 and 132 complex points in the ¹H and ¹³C dimensions, respectively, with spectral widths of 12000 Hz and 4100 Hz. 16 scans were obtained per individual FID.

### NMR acquisition of three-dimensional NOESY spectra

The 3D HSQC-NOESY-HSQC NMR experiment on MSG was performed on a ~ 400 μM MSG sample on a Bruker 950 MHz Avance HD spectrometer equipped with Z-gradient triple-resonance TCI cryoprobe at 310 K. The data was acquired with 1024, 142, and 124 complex points in ¹H, ¹³C_HSQC, and ¹³C_NOESY dimensions, respectively, with spectral widths of 15244 Hz (¹H), 6667 Hz (¹³C), and 6667 Hz (¹³C). Eight scans were collected per increment with a recycle delay of 1 s. The mixing time was 60 ms.

### NMR data processing

All experimental NMR spectra were processed with NMRPIPE[42] version 2018.184.13.26 or using the python libraries NMRGLUE[43] 0.9 and NUMPY 1.23.5 and visualised/analysed with CARA 1.8.4.2 (http://cara.nmr.ch/) and NMRDRAW 2018.184.13.26.

### Reporting summary

Further information on research design is available in the Nature Portfolio Reporting Summary linked to this article.

## Data availability

Training data for the two DNNs generated in this study have been deposited at Zenodo (www.zenodo.org), Hansen, D. F. (2024) "Solution-State Methyl NMR Spectroscopy of Large Non-Deuterated Proteins Enabled by Deep Neural Networks". *Zenodo*. https://doi.org/10.5281/zenodo.10022405. Synthetic spectra used for assessment of FID-Net are available as PDF files. The full spectra used for experimental assessments, including, (*i*) input ¹³C-¹H HSQC, (*ii*) FID-Net processed ¹³C-¹H HSQC, and (*iii*) methyl-TROSY for HDAC8, MSG, and α7α7-proteasome are also available from Zenodo, along with the full ¹³C-¹³C-¹H NOESY spectrum of MSG. There reference PDB code used in this work is 1D8C. Source data are provided with this paper.

## Code availability

Python code for using the networks described here (including pre-trained networks, examples and scripts for training the two DNNs) is available on GitHub: https://github.com/gogulan-k/FID-Net. Training scripts are available from Zenodo: https://doi.org/10.5281/zenodo.11080581[44].

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

## Acknowledgements

The BBSRC (BB/R000255/1; D.F.H.), Wellcome Trust (ref. 101569/z/13/z; D.F.H.), and the EPSRC are acknowledged for supporting the NMR facility at University College London. Access to ultra-high field NMR spectrometers was supported by the Francis Crick Institute through

provision of access to the MRC Biomedical NMR Centre and by the University of Oxford Wellcome Institutional Strategic Support Fund, the John Fell Fund, as well as the Edward Penley Abraham Cephalosporin Fund, and the Engineering and Physical Sciences Research Council (EP/R029849/1). The Francis Crick Institute receives its core funding from Cancer Research UK (FC001029), the UK Medical Research Council (FC001029), and the Wellcome Trust (FC001029). This study made use of NMRbox: National Center for Biomolecular NMR Data Processing and Analysis, a Biomedical Technology Research Resource (BTRR), which is supported by NIH grant P41GM111135 (NIGMS). Some computational aspects of this work were supported by the Francis Crick Institute (D.F.H.) through provision of access to the Scientific Computing STP and the Crick data Analysis and Management Platform (CAMP). The Francis Crick Institute (CAMP) receives its core funding from Cancer Research UK (FC010233), the UK Medical Research Council (FC010233), and the Wellcome Trust (FC010233). For the purpose of open access, the author has applied a Creative Commons Attribution (CC BY) licence to any Author Accepted Manuscript version arising. This research is supported by the UKRI and EPSRC (EP/X036782/1; D.F.H.).

## Author contributions

G.K. designed and trained all the DNNs, G.K. and V.K.S. produced iso-tope labelled samples; G.K., V.K.S., and D.F.H. performed NMR experiments; V.K.S assigned the chemical shifts of methyl spectrum. G.K. and D.F.H designed the research. All authors analysed the data, discussed the results, and wrote the paper.

## Competing interests

The authors declare no competing interests.
