## [Peer Review File · Nature Communications]

Solution-State Methyl NMR Spectroscopy of Large Non-Deuterated Proteins Enabled by Deep Neural NetworksREVIEWER COMMENTS

Reviewer #1 (Remarks to the Author):

The paper by Karunanithy et al. describes an important and innovative application of Deep Neural Networks (DNN) to Methyl NMR Spectroscopy of high-molecular-weight proteins. The work is groundbreaking and has a potential to find a wide application for bio-molecular NMR research in general and methyl-based NMR of large proteins in particular. I highly recommend it for publications in Nature Communications after a number of relatively minor issues are addressed by the authors.

Apart from quite extensive copy-editing that should be performed by the authors (multiple instances of plurals employed instead singular forms of verbs/nouns and vice versa; numerous instances of the wrong word ordering and confusing phrasings etc etc), I would recommend addressing the issue of the range of applicability of the described DNN processing algorithms in some more detail, as it seems to be a bit 'garbled' in the current version of the paper. On page 16 ('Discussion' section) the authors mention that the main shortcoming of the FID-net methodology is "that the process of peak sharpening inevitably leads to the intrinsic peak intensity being lost".

Do the authors envisage some (future) remedy for this important drawback that makes FID-net inapplicable to any quantitative NMR applications ??

Along the same lines, are the limits on molecular weight ranges for which FID-net can be employed (put by the authors at the half-proteasome size (~360 kD) on pp.12-13 and in Fig. 4) dictated by the same considerations (i.e. problems with 'peak-sharpening' algorithm in the 1H dimension of NMR spectra ??

Minor points:

1. I did not understand what grey circles in Fig. 5f represent and how different they are from what is shown by blue circles. Please re-phrase/revise the cryptic explanation of what is actually plotted in this figure in the legend and in the text.

2. The authors refer on a couple of occasions to a "1822 Fourier transform". Is this any different from a standard "Fourier transform" ?

If the intention of the authors is to remind the reader that Fourier transform dates back to 1822 (which is probably the case), then

I would argue that the algorithms for Fourier transform that are currently used in NMR spectroscopy are all "Fast Fourier Transform"

and are much newer.....

3. Several papers (published mostly in the Journal of Biomolecular NMR) described assignments of Ala beta, Ile gamma 2, and Threonine gamma methyl groups in MSG.

I recommend that the authors find these papers and compare their NOE-based assignments of these methyl sites to the published ones and mention the results of this comparison in the revised manuscript.

Reviewer #2 (Remarks to the Author):

Methyl TROSY is an important and commonly used technique in NMR spectroscopy that allows characterization of large biomolecules. In this study, the authors propose a novel approach that leverages deep learning to enhance the quality of the NMR spectra and enable the measurement without sample deuteration.

The proposed approach is original, following the previous work of the authors on FID-Net, it aligns with current trends of utilizing deep learning in macromolecular NMR spectroscopy, and offers a practical solution for NMR practitioners. However, the manuscript would benefit from more detailed evaluation of the proposed technique, allowing the readers to assess its strengths and limitations.

MAJOR COMMENTS:

1. One of the main potential benefits of the method is to simplify the analysis of the spectra fragments that contain highly overlapping peaks, as their manual analysis is difficult. On the contrary, isolated peaks are easier to identify, sometimes being identifiable even in spectrum that constitutes input for FID-Net (e.g. Figure 2A). In line 165 the authors make a statement that only peaks that are isolated in the processed spectra are considered for quantitative analysis. Presumably this should be further clarified. What is the average number of isolated peaks used in quantitative analysis, compared to total number of expected peaks in the spectrum? What is the true positives rate and false negatives rate when calculated for all peaks?

2. True positives and false negative rates are reported (e.g. Figure 2). However, I'm wondering why

false positives are not reported as the third quantity, what is common practice in machine learning projects. Is it possible for the FID-Net to introduce artificial signals into the spectrum that could be interpreted as true peak (e.g., by a peak picking method)?

3. Training machine learning models on a synthetic dataset is common and frequently unavoidable. However, machine learning models trained on synthetic datasets typically exhibit performance gap between synthetic test set and real data. The manuscript provides quantification of the network errors on the synthetic test set (true positives, false negatives), but similar quantification is missing for experimental data.

4. FID-Net was trained with mean squared error (MSE) in the frequency domain. In principle, this metric can complement quantitative evaluation that involves peak picking. It can also be applied to the whole spectrum, not just to isolated peaks. Regarding the simulated validation set, one could verify the ratio of MSE between the model output and the ground truth, compared to the MSE between the model input and the ground truth?

MINOR COMMENTS:

1. Line 67. ARTINA is one of the large deep learning projects for the analysis of complex NMR data. The method, published in 2022, was made publicly available through NMRtist platform (<https://nmrtist.org/>) and have processed almost 30 000 jobs submitted by the users. The authors may consider referring to this method in order to show more complete landscape of the recent work in the field.

Klukowski, P., Riek, R. & Güntert, P. Rapid protein assignments and structures from raw NMR spectra with the deep learning technique ARTINA (2022). *Nature Communications* 13, 6151.

2. Line 78. This statement suggests that the proposed method could entirely replace Fourier transform (e.g. the FID-Net takes the signal in time domain, and outputs enhanced signal in frequency domain). Later (line 145) it is explained that DNN output is Fourier Transformed. Presumably, one could clarify the statement: “we demonstrate that deep neural networks (DNNs) can be used, in place of the traditional 1822 Fourier transform, to deliver very high-quality ¹³C-¹H correlation spectra”.

3. In lines 163 and 164, as well as in the caption of Figure 2, the authors refer to “true positives” and “false negatives”. Figure 2 presents “true positives” and “false positives”. Should the text in figure be changed to “true positives” and “false negatives” to maintain the consistency with the text?

4. The authors use the keyword “cross-validation” throughout the manuscript (e.g. lines 207, 415, 424). In computer science “cross-validation” refers to resampling method that uses different portions of the data to test and train a model on different iterations (e.g. K-fold cross-validation, one-leave-out cross-validation). Was cross-validation experiment carried out in this study, or the validation of the model was made through splitting the dataset into training and testing sets?

5. The link in “data availability” section points to the home page of Zenodo, not the specific dataset associated with the manuscript.

Reviewer #3 (Remarks to the Author):

This is an excellent paper that convincingly demonstrates the use of deep neural networks to sharpen up methyl 1H - ^{13}C cross-peaks in a fully protonated sample of a large (by NMR standards) protein, namely the 82 kDa MSG.

The work is solid and well presented and I have no suggestions for improvement. There is also no question that the approach is ingenious and provides a very nice example of the power of deep neural networks when applied intelligently (as in this instance).

That being said, it does seem to me that in some sense, while sophisticated in terms of data processing, the method presented represents a poor man's method in so far that perdeuteration and methyl group labeling (using ILV but Met and Ala as well) offers far more quantitative spectral data (which is essential for example, for any sort of relaxation-based studies (including the use of relaxation dispersion to study excited states). In that regard the argument that methyl labeling and perdeuteration is excessively expensive is perhaps a little off and should probably be toned down given that this is standard practice when studying proteins of this size. Of course it is also perfectly true that there are systems where expression in *E. coli* is not feasible and perdeuteration and methyl labeling becomes difficult.

In conclusion I recommend acceptance in *Nat. Comm* with very minor revision of the Discussion and Conclusion sections.

Addressing Reviewers comments

Reviewer #1:

The paper by Karunanithy et al. describes an important and innovative application of Deep Neural Networks (DNN) to Methyl NMR Spectroscopy of high-molecular-weight proteins. The work is groundbreaking and has a potential to find a wide application for bio-molecular NMR research in general and methyl-based NMR of large proteins in particular. I highly recommend it for publications in Nature Communications after a number of relatively minor issues are addressed by the authors.

We are delighted with the favourable comments made by reviewer 1 on our manuscript. Below is a point-by-point response to the specific comments, **where the reviewer's comments are in red**, our responses are in black, and new text included in the revised manuscript or in the revised supporting material is **highlighted in yellow**.

A) Apart from quite extensive copy-editing that should be performed by the authors (multiple instances of plurals employed instead singular forms of verbs/nouns and vice versa; numerous instances of the wrong word ordering and confusing phrasings etc etc)

Response:

We have carefully gone through the manuscript and also had other native English and American speakers proofread the manuscript.

B) I would recommend addressing the issue of the range of applicability of the described DNN processing algorithms in some more detail, as it seems to be a bit 'garbled' in the current version of the paper. On page 16 ('Discussion' section) the authors mention that the main shortcoming of the FID-net methodology is "that the process of peak sharpening inevitably leads to the intrinsic peak intensity being lost".

Response:

We believe that the main use of the presented strategy is for chemical shift based studies e.g. to understand both intrinsic structure and changes due to different conditions or addition of an interaction partner. The FID-Net methodology could also be used as a useful tool in the assignment of large proteins, where point mutations are often employed. The loss of quantitative information on the shape of the cross-peaks means it is not useful for line-shape analysis and we also do not advise using the methodology in studies where the peak intensity is crucial e.g. relaxation or diffusion studies. We have now added more details to the section in the discussion section:

“The main disadvantage of the FID-Net method is that the process of peak-sharpening inevitably leads to a loss of the intrinsic shape of the cross-peaks, including the peak height. Accurately measuring peak intensities is critical in a number of NMR experiments, including diffusion and relaxation, so it is not advised to record these experiments in conjunction with the presented FID-Net processing nor is it advised to perform line-shape analyses. However, for a large body of NMR experiments, the main parameter of importance is the chemical shift as well as a reasonable estimate of the peak intensity, and in these cases we believe that FID-Net processing will prove extremely useful. Such studies include classical chemical shift perturbation studies, such as, changes in chemical shifts upon addition of an interacting partner

or changes related to varying conditions, e.g. pH. As we have shown above, the obtained intensities are of a sufficient quality such that the transformed spectra facilitate facile chemical shift assignment of methyl peaks by either NOESY spectra (as demonstrated here for MSG) or by point mutations, which often requires several samples.”

C) Do the authors envisage some (future) remedy for this important drawback that makes FID-net inapplicable to any quantitative NMR applications ??

Response:

The reviewer brings up a very important point, which is indeed a key part of our current research. Briefly, in order to make the current FID-network quantitative, we would need to retrain the network with additional constraints or one could reverse the tangent-hyperbolic operation, although the network would then lose its peak-sharpening abilities. We are currently training a new architecture to do a similar task to the FID-net methodology presented above, however, where quantitative peak volumes (and their uncertainties) can be obtained. This has required a redesign of the FID-Net architecture and this new network is still being trained. The new network will be published in due course and a description and discussion of this will be out of scope for the current manuscript. We have added the following to the discussion:

“We are currently in the process of developing and training a DNN that is able to enhance the resolution of ^{13}C - ^1H spectra and provide quantitative mappings, which can be used for downstream analyses, e.g. relaxation experiments. This work remains ongoing.”

D) Along the same lines, are the limits on molecular weight ranges for which FID-net can be employed (put by the authors at the half-proteasome size (~360 kD) on pp.12-13 and in Fig. 4) dictated by the same considerations (i.e. problems with 'peak-sharpening' algorithm in the 1H dimension of NMR spectra ??

Response:

The size limits of this technique arise due to the fact that the DNN is reliant in some signal being present in the data so that it can be sharpened. Beyond 360 kDa we find there is very limited signal in the first place so the methodology does not work. We have added the following sentence to the discussion:

“The size limitation for FID-Net based processing arises because it is reliant on some signal still being present in the spectra and above this size most signals, even those associated with methyl moieties, are broadened beyond detection in non-deuterated systems.”

Minor points:

1. I did not understand what grey circles in Fig. 5f represent and how different they are from what is shown by blue circles. Please re-phrase/revise the cryptic explanation of what is actually plotted in this figure in the legend and in the text.

Response:

We have now updated the figure legend to make it clearer, what the difference is between the grey and blue circles.

(f) Normalised NOE cross-peak volumes (cross-peak volume/diagonal-peak volume) versus interproton distances. Grey circles represent the normalised NOE cross-peak volumes obtained for each individual NOE cross-peaks, whereas blue circles represent the average of normalised NOE cross-peaks volume over interproton-distance intervals of 0.2 Å, i.e. (Sum of normalised NOE cross-peak volumes) / (Number of cross-peaks), within each interproton-distance intervals of 0.2 Å. The blue line represents the fitted curve of NOE cross-peaks volume (V) and interproton distance (r) using the standard equation $V = C/r^6$, where C is a constant.

2. The authors refer on a couple of occasions to a "1822 Fourier transform". Is this any different from a standard "Fourier transform" ? If the intention of the authors is to remind the reader that Fourier transform dates back to 1822 (which is probably the case), then I would argue that the algorithms for Fourier transform that are currently used in NMR spectroscopy are all "Fast Fourier Transform" and are much newer....

Response:

We felt originally that it would be of interest to stress to the readers that the classical Fourier Transform, albeit an utmost important algorithm and mapping, was developed more than 200 years ago. It might therefore be timely to look towards new mappings / transformations of NMR spectra. Based on the reviewers' comments, we do acknowledge that this might have caused some confusion. We have therefore removed '1822' in the revised manuscript and just refer to this as the 'Fourier transform' or 'fast Fourier transform'

3. Several papers (published mostly in the Journal of Biomolecular NMR) described assignments of Ala beta, Ile gamma 2, and Threonine gamma methyl groups in MSG. I recommend that the authors find these papers and compare their NOE-based assignments of these methyl sites to the published ones and mention the results of this comparison in the revised manuscript.

Response:

We have added all these references in the manuscript and also compared the published assignments of the methyl groups with our spectra. In the FID-Net processed spectrum of MSG, peaks for all $^{13}\text{C}^{\gamma 2}$ methyl groups of Ile and Thr, $^{13}\text{C}^{\epsilon}$ of Met, and $^{13}\text{C}^{\beta}$ of Ala were observed, except for $^{13}\text{C}^{\beta}$ of A633. We have added following sentences in the manuscript.

“ The FID-Net processed spectrum of MSG also agrees with the previously published chemical shift assignments for $^{13}\text{C}^{\gamma 2}$ methyl groups of Ile and Thr, $^{13}\text{C}^{\epsilon}$ of Met, and $^{13}\text{C}^{\beta}$ of Ala²⁶⁻²⁸, with the exception of $^{13}\text{C}^{\beta}$ of A633, which was not observed in the FID-Net processed spectra.”

Reviewer #2:

... The proposed approach is original, following the previous work of the authors on FID-Net, it aligns with current trends of utilizing deep learning in macromolecular NMR spectroscopy, and offers a practical solution for NMR practitioners. However, the manuscript would benefit from more detailed evaluation of the proposed technique, allowing the readers to assess its strengths and limitations.

We are delighted that reviewer 2 find our work and approach original. Below is a point-by-point response to the specific comments, where the reviewer's comments are in red, our responses are in black, and new text included in the revised manuscript or in the revised supporting material is highlighted in yellow.

MAJOR COMMENTS:

1. One of the main potential benefits of the method is to simplify the analysis of the spectra fragments that contain highly overlapping peaks, as their manual analysis is difficult. On the contrary, isolated peaks are easier to identify, sometimes being identifiable even in spectrum that constitutes input for FID-Net (e.g. Figure 2A). In line 165 the authors make a statement that only peaks that are isolated in the processed spectra are considered for quantitative analysis. Presumably this should be further clarified. What is the average number of isolated peaks used in quantitative analysis, compared to total number of expected peaks in the spectrum? What is the true positives rate and false negatives rate when calculated for all peaks?

Response:

We used only isolated peaks to ensure that the assessment was mainly driven by the effect of FID-Net rather than the quality of peak picking. We have added a supplementary file to show all the peaks that have been used in the analysis for both the HDAC-like and MSG-like spectra.

We also note that performing an analysis on all the peaks in the transformed spectra is more difficult since the FID-Net approach will only remove a single coupling from the peaks. Thus, while methyl moieties will be mapped into singlets, triplets due to -CH₂- groups will be mapped into doublets. We believe this has some advantage in allowing methyl peaks to be easily identified in the final transformed spectra. By providing all the spectra for which we assessed FID-Net, we hope it is clear by inspection that even more crowded regions of the spectra are treated well. We now also show a more quantitative analysis using RMSE error below and also test the performance of FID-Net on real spectra including crowded regions in Figure 3.

When we include all the peaks in the spectra using the simulated target spectra where all the peaks are singlets for the MSG-like spectra we find that the true positive rate falls to 88.9%, the false negative rate is 11.1% and the false positive rate is 0.9% showing the fallibility of the peak picker for crowded spectra even in idealised cases.

2. True positives and false negative rates are reported (e.g. Figure 2). However, I'm wondering why false positives are not reported as the third quantity, what is common practice in machine learning projects. Is it possible for the FID-Net to introduce artificial signals into the spectrum that could be interpreted as true peak (e.g., by a peak picking method)?

Response:

There was previously an error in the text and the values reported in Figure 2 (as stated in the figure) were true positives and false positives. We have now updated the text as below and also added false negatives as an additional quantity to figure 2.

“We then pick peaks in the resulting transformed spectra and compare the results against the known peak positions by quantifying the rate of true positives, false positives, and false negatives.”

3. Training machine learning models on a synthetic dataset is common and frequently unavoidable. However, machine learning models trained on synthetic datasets typically exhibit performance gap between synthetic test set and real data. The manuscript provides quantification of the network errors on the synthetic test set (true positives, false negatives), but similar quantification is missing for experimental data.

Response:

We agree with the reviewer that we simply do not have enough experimental data for training the DNN on experimental data. Our approach is to use the experimental data that we have purely for cross-validations. These experimental data will then truly provide excellent validations because they are real data and allow us to see how the DNN performs in a real experimental scenario.

However, we cannot know the ground truth for experimental data, which therefore somewhat limits the cross-validations. What we have done is therefore to provide overlays of experimental data (deuterated methyl-TROSY spectra) v.s. the outputs from the DNN (Figures 3 and 4). We have now additionally compared previously published chemical shift assignments for methyl groups in MSG and see that all but one assignments are visible in the FID-Net processed spectra. Thus, for both HDAC8 and MSG, all the cross-peaks for assigned methyl groups of HDAC8 and MSG are present in the FID-Net processed spectrum, except for $^{13}\text{C}^\beta$ of A633 of MSG. We have added the following:

“All previously assigned methyl cross-peaks for HDAC8²⁷ were present in the FID-Net processed spectra of HDAC8”.

and

“The FID-Net processed spectrum of MSG also agrees with the previously published chemical shift assignments for $^{13}\text{C}^{\gamma 2}$ methyl groups of Ile and Thr, $^{13}\text{C}^\epsilon$ of Met, and $^{13}\text{C}^\beta$ of Ala²⁶⁻²⁸, with the exception of $^{13}\text{C}^\beta$ of A633, which was not observed in the FID-Net processed spectra.”

4. FID-Net was trained with mean squared error (MSE) in the frequency domain. In principle, this metric can complement quantitative evaluation that involves peak picking. It can also be applied to the whole spectrum, not just to isolated peaks. Regarding the simulated validation

set, one could verify the ratio of MSE between the model output and the ground truth, compared to the MSE between the model input and the ground truth?

Response:

We have compared the RMSE between the target and transformed spectra for both HDAC and MSG-like spectra. In both cases we only consider regions where we get singlet peaks to avoid RMSE being inflated by inaccuracy in the peak-picking and doublets in the spectra. Here we find for the HDAC-like spectra the average RMSE over one hundred synthetic spectra is 0.0501 and for MSG-like spectra it is 0.0580. We have attached files showing the regions over which the RMSE is calculated.

MINOR COMMENTS:

1. Line 67. ARTINA is one of the large deep learning projects for the analysis of complex NMR data. The method, published in 2022, was made publicly available through NMRtist platform (<https://nmrtist.org/>) and have processed almost 30 000 jobs submitted by the users. The authors may consider referring to this method in order to show more complete landscape of the recent work in the field.

Klukowski, P., Riek, R. & Güntert, P. Rapid protein assignments and structures from raw NMR spectra with the deep learning technique ARTINA (2022). *Nature Communications* 13, 6151.

Response:

We thank the reviewer for pointing out this article, which certainly is relevant to reference in the current manuscript. We have now added a reference to the ARTINA article.

2. Line 78. This statement suggests that the proposed method could entirely replace Fourier transform (e.g. the FID-Net takes the signal in time domain, and outputs enhanced signal in frequency domain). Later (line 145) it is explained that DNN output is Fourier Transformed. Presumably, one could clarify the statement: “we demonstrate that deep neural networks (DNNs) can be used, in place of the traditional 1822 Fourier transform, to deliver very high-quality ^{13}C - ^1H correlation spectra”.

Response:

We have now changed this to:

“Herein, we demonstrate that deep neural networks (DNNs) can be used, in conjunction with the traditional Fourier transform¹⁶, to deliver very high-quality ^{13}C - ^1H correlation spectra from uniformly ^{13}C protonated samples, including large proteins whose size limits have traditionally rendered them inaccessible to NMR.”

3. In lines 163 and 164, as well as in the caption of Figure 2, the authors refer to “true positives” and “false negatives”. Figure 2 presents “true positives” and “false positives”. Should the text in figure be changed to “true positives” and “false negatives” to maintain the consistency with the text?

Response:

The text was previously incorrect in saying false negatives rather than false positives. We have now updated the figure and text to explicitly include true positives, false positives and false negatives.

“We then pick peaks in the resulting transformed spectra and compare the results against the known peak positions by quantifying the rate of true positives, false positives, and false negatives.”

4. The authors use the keyword “cross-validation” throughout the manuscript (e.g. lines 207, 415, 424). In computer science “cross-validation” refers to resampling method that uses different portions of the data to test and train a model on different iterations (e.g. K-fold cross-validation, one-leave-out cross-validation). Was cross-validation experiment carried out in this study, or the validation of the model was made through splitting the dataset into training and testing sets?

We thank the reviewer for pointing this out. Throughout the manuscript the word cross-validation has been replaced with more appropriate terminology.

5. The link in “data availability” section points to the home page of Zenodo, not the specific dataset associated with the manuscript.

Response:

The data were already uploaded to Zenodo, and they are now published. The link has been updated. Training scripts have also been added to the GitHub repository

Reviewer #3:

This is an excellent paper that convincingly demonstrates the use of deep neural networks to sharpen up methyl $1\text{H}-13\text{C}$ cross-peaks in a fully protonated sample of a large (by NMR standards protein, namely the 82 kDa MSG).

The work is solid and well presented and I have no suggestions for improvement. There is also no question that the approach is ingenious and provides a very nice example of the power of deep neural networks when applied intelligently (as in this instance).

In conclusion I recommend acceptance in Nat. Comm with very minor revision of the Discussion and Conclusion sections.

We are delighted that reviewer 3 find our work and work excellent, solid, well-presented and convincing. Below response to the specific comment raised by this reviewer, **where the reviewer's comments are in red**, our responses are in black, and new text included in the revised manuscript or in the revised supporting material is **highlighted in yellow**.

1) That being said, it does seem to me that in some sense, while sophisticated in terms of data processing, the method presented represents a poor man's method in so far that perdeuteration and methyl group labeling (using ILV but Met and Ala as well) offers far more quantitative spectral data (which is essential for example, for any sort of relaxation-based studies (including the use of relaxation dispersion to study excited states). In that regard the argument that methyl labeling and perdeuteration is excessively expensive is perhaps a little off and should probably be toned down given that this is standard practice when studying proteins of this size. Of course it is also perfectly true that there are systems where expression in *E. coli* is not feasible and perdeuteration and methyl labeling becomes difficult.

Response:

We have now toned down the references to cost and emphasise the aspect that deuteration and that methyl labelling in a deuterated background is often not feasible for systems that cannot be expressed in bacterial expression systems such as *E. Coli*. Please see main changes below. We have kept the cost calculated based on listed prices in Figure 1, because we believe that provides an accurate, non-biased, representation of the difference in cost.

In the abstract, we now focus on the potential ability to characterise proteins that cannot be produced in *E. coli*:

“... potentially providing information on proteins that cannot be produced in bacterial systems.”

In the introduction we removed ‘considerable’ and now write:

“However, this uniform deuteration has several disadvantages, including extra costs and typically lower yields of expressed protein. Furthermore, deuteration is not possible for many systems of considerable biological interest, including proteins that can only be expressed in mammalian systems.”

In the result section we have removed:

“... being far less costly than deuterated analogues“ and changed “at a fraction of the cost” with “at a lower cost”

REVIEWERS' COMMENTS

Reviewer #1 (Remarks to the Author):

I think the paper can be published in its present form.
The authors have resolved in the revised version of the text
all the raised issues/concerns to my full satisfaction.

Reviewer #2 (Remarks to the Author):

I would like to thank the authors for thoroughly addressing my comments. The responses provided are convincing, and the modifications made to the manuscript positively enhance its readability and overall quality. This work represents an advancement that is likely to have a notable impact in the field.

I don't have any further comments and recommend the work for publication in Nature Communications in its current form.